# Optimal Sketching for Residual Error Estimation for Matrix and Vector Norms

**Yi Li**
Nanyang Technological University
yili@ntu.edu.sg

**Honghao Lin, David P. Woodruff**
Carnegie Mellon University
{honghaol, dwoodruf}@andrew.cmu.edu

## Abstract

We study the problem of residual error estimation for matrix and vector norms using a linear sketch. Such estimates can be used, for example, to quickly assess how useful a more expensive low-rank approximation computation will be. The matrix case concerns the Frobenius norm and the task is to approximate the $k$-residual $\|A - A_k\|_F$ of the input matrix $A$ within a $(1 + \varepsilon)$-factor, where $A_k$ is the optimal rank-$k$ approximation. We provide a tight bound of $\Theta(k^2/\varepsilon^4)$ on the size of bilinear sketches, which have the form of a matrix product $SAT$. This improves the previous $O(k^2/\varepsilon^6)$ upper bound in (Andoni et al. SODA 2013) and gives the first non-trivial lower bound, to the best of our knowledge. In our algorithm, our sketching matrices $S$ and $T$ can both be sparse matrices, allowing for a very fast update time. We demonstrate that this gives a substantial advantage empirically, for roughly the same sketch size and accuracy as in previous work.

For the vector case, we consider the $\ell_p$-norm for $p > 2$, where the task is to approximate the $k$-residual $\|x - x_k\|_p$ up to a constant factor, where $x_k$ is the optimal $k$-sparse approximation to $x$. Such vector norms are frequently studied in the data stream literature and are useful for finding frequent items or so-called heavy hitters. We establish an upper bound of $O(k^{2/p} n^{1-2/p} \operatorname{poly}(\log n))$ for constant $\varepsilon$ on the dimension of a linear sketch for this problem. Our algorithm can be extended to the $\ell_p$ sparse recovery problem with the same sketching dimension, which seems to be the first such bound for $p > 2$. We also show an $\Omega(k^{2/p} n^{1-2/p})$ lower bound for the sparse recovery problem, which is tight up to a $\operatorname{poly}(\log n)$ factor.

## 1 Introduction

Low-rank approximation is a fundamental task for which, given an $m \times n$ matrix $A$, one computes a rank-$k$ matrix $B$ for which $\|A - B\|_F^2$ is small. This works well in practice since it is often the case that matrices are close to being low rank, and only have large rank because of a small amount of noise. Also, $B$ provides a significant compression of $A$, involving only $(m + n)k$ parameters (if $B$ is represented in factored form) rather than $mn$, which then makes it quicker to compute matrix-vector products and so on.

However, computing a low-rank approximation can be expensive if $m$ and $n$ are large. While there exist fast sketching techniques, see, e.g., (Woodruff, 2014a), such techniques would still require $\Omega(k(m + n))$ memory even to write down the output $B$ in factored form, and a stronger lower bound of $\Omega(k(m + n)/\varepsilon)$ exists in the data stream model (Clarkson & Woodruff, 2009), even if the rows or columns appear in order (Woodruff, 2014b). Here $\varepsilon$ is the desired accuracy, so one should have $\|A - B\|_F \le (1 + \varepsilon)\|A - A_k\|_F$, where $A_k$ is the best rank-$k$ approximation to $A$ in Frobenius norm.

Given that it is expensive both in time and memory to compute a low-rank approximation, it is natural to first ask if there is value in doing so. We would like $\|A - B\|_F$ to be as small as possible, and much less than $\|A\|_F$ so that $B$ accurately represents $A$. One way to do this is to try to first estimate the *residual error* of $\|A - A_k\|_F$, which potentially one can do with an amount of memory depending only on $k$ and $\varepsilon$. Indeed, this is precisely what (Andoni & Nguyen, 2013) show, namely, that by computing a sketch $SAT$, where $S$ and $T$ are random Gaussian matrices each with small dimension $O(k/\varepsilon^3)$, ignoring logarithmic factors, that $\|(SAT) - (SAT)_k\|_F = (1 \pm \varepsilon)\|A - A_k\|_F$. Note that it is important that $S$ and $T$ do not depend on $A$ itself, as one may not have access to the parts of

$A$ in a stream needed when applying $S$ and $T$, or $A$ may be distributed across multiple servers, etc. That is, $S$ and $T$ are said to be *oblivious* sketches, which is the focus of this paper. Thus, with only $O(k^2/\varepsilon^6)$ memory words in a stream, one can first figure out if it is worthwhile to later compute a low-rank approximation to $A$.

There are several weaknesses to the above result. First, it is unclear if the $O(k/\varepsilon^3)$ dimension bounds are tight, and optimizing them is especially important for small $\varepsilon$. Can the upper bound be improved? Also, the only known lower bound on the small dimension of sketches $S$ and $T$ and even when one requires an estimator of the form $\|(SAT) - (SAT)_k\|_F$, as far as we are aware, is $\Omega(k + 1/\varepsilon^2)$. This follows from both $S$ and $T$ needing to preserve rank and preserve the norm of a fixed vector. Such a lower bound is even more unclear if we are allowed an arbitrary recovery procedure $f(S, T, SAT)$. Second, the running time is not ideal, as each entry of $A$ in the stream needs to be multiplied on the left and right by dense Gaussian sketches.

Residual error is not only a useful concept for matrices, but can capture how useful a sparse approximation is for vectors, which is the standard way of compressing a vector using compressed sensing. In this case, one would like to compute a sketch $S \cdot x$ of an underlying $n$-dimensional vector $x$ so that one can output a $k$-sparse vector $\hat{x}$ for which $\|x - \hat{x}\|_p \leq (1 + \varepsilon)\|x - x_k\|_p$, where $x_k$ is the best $k$-sparse approximation to $x$, namely, the vector formed by taking the $k$ coordinates of $x$ of the largest absolute value. Here $\|z\|_p = (\sum_{i=1}^n |z_i|^p)^{1/p}$ is the vector $p$-norm. Just like for low-rank approximation, one could ask if it is worthwhile to compute $\hat{x}$, and that can be determined based on whether $\|x - x_k\|_p$ is small. This is also useful in the data stream literature, and is referred to as the heavy hitters estimation problem with tail error, and $p > 2$ enables to find heavy hitters that are even "less heavy" than those for $\ell_2$, see, e.g., (Berinde et al., 2010), or residual heavy hitters, see, e.g., Section 4 of (Harvey et al., 2008) for applications to entropy estimation in a stream as well as (Indyk, 2004) for applications to computational geometry in a stream.

For $1 \leq p \leq 2$ there are very efficient sketches to compute $\hat{x}$ itself, which are sparse and only involve a sketching dimension of $O(k/\varepsilon^2)$ up to logarithmic factors (Price & Woodruff, 2011). Notably, for the $\ell_p$ sparse recovery problem where $p \in \{1, 2\}$ and we are required to output a vector $\hat{x}$, an $\Omega(k \log(n/k))$ lower bound on the sketching dimension holds (Ba et al., 2010; Price & Woodruff, 2011), while if we only need to estimate $\|x - x_k\|_p$, following the idea of (Indyk et al., 2011) where we first perform a dimensionality reduction to $\mathrm{poly}(k/\varepsilon)$ dimensions, one can show that $O(k \log(k))$ measurements suffice, which shows a separation between the two problems. Motivated by this, we focus on residual norm estimation for $p > 2$, which is a choice of $p$ that has been the focus of a long line of work on frequency moments in a data stream, starting with (Alon et al., 1999). For such $p$, it is not even known what the right dependence on $n$ and $k$ is, so we focus on constant $\varepsilon$ for this problem.

## 1.1 Our Contributions

For residual norm estimation for low-rank approximation by sketches of the form $SAT$ and estimators of the form $\|SAT - [SAT]_k\|_F$, we improve the bound of (Andoni & Nguyen, 2013), showing that both $S$ and $T$ can have small dimension $O(k/\varepsilon^2)$, up to logarithmic factors, rather than $O(k/\varepsilon^3)$. Moreover, our sketch can be the composition of a CountSketch and a Gaussian matrix (Clarkson & Woodruff, 2013), or use OSNAP (Nelson & Nguyen, 2012) as analyzed by Cohen (Cohen, 2016) to achieve faster runtime or update time in a stream. We complement this upper bound with a matching lower bound for bilinear sketches $SAT$ and an arbitrary recovery procedure $f(S, T, SAT)$, where we show that both $S$ and $T$ need to have $\Omega(k/\varepsilon^2)$ small dimension, matching our upper bound.

For the residual vector norm estimation problem, we in fact design the first sparse recovery algorithms, i.e., compressed sensing algorithms, for $p > 2$, designing a sketching matrix $S$ with $n^{1-2/p}k^{2/p}\,\mathrm{poly}(\log n)$ small dimension for recovering the vector $\hat{x}$ itself. By running a standard sketching algorithm for $p$-norm estimation in parallel, we can thus estimate $\|x - \hat{x}\|_p$ to evaluate the residual cost. We show that at least for the sparse recovery problem there is a nearly matching lower bound on the sketching dimension of $n^{1-2/p}k^{2/p}$. While we do not resolve the sketching dimension of residual norm estimation, a lower bound of $n^{1-2/p}$ follows from previous work (Bar-Yossef et al., 2004) and so a polynomial dependence on $n$ is required.

Finally, we empirically evaluate our residual norm estimation algorithm for low-rank approximation on real data sets, showing that while we achieve similar error for the same sketching dimension, our sketch is 4 to 7 times faster to evaluate.

## 2 PRELIMINARIES

**Notation.** For an $n \times d$ matrix $A$, let $A_k$ denote the best rank-$k$ approximation of $A$ and $\sigma_1(A) \geq \sigma_2(A) \geq \ldots \geq \sigma_s(A)$ denote its singular values where $s = \min\{n, d\}$. We then have $\|A - A_k\|_F^2 = \sum_{i=k+1}^s \sigma_i^2$. Given a vector $x \in \mathbb{R}^n$, let $\|x\|_p = (\sum_i |x_i|^p)^{1/p}$ denote the $p$-norm of the vector $x$. Let $x_k \in \mathbb{R}^n$ denote the best $k$-sparse vector approximation to $x$, such that we keep the top $k$ coordinates of $x$ in absolute value and make the remaining coordinates 0. Let $x_{-k} = x - x_k$. We next formally define the problems and models we consider.

**Low-Rank Residual Error Estimation.** Given a matrix $A \in \mathbb{R}^{n \times d}$, our goal is to estimate the rank-$k$ residual error $\|A - A_k\|_F$ within a $(1 \pm \varepsilon)$ multiplicative factor, where $A_k$ is the best rank-$k$ approximation to $A$.

**Residual Vector Norm Estimation.** In the data stream model, we assume there is an underlying *frequency vector* $x \in \mathbb{Z}^n$, initialized to $0^n$, which evolves throughout the course of a stream. The stream consists of updates of the form $(i, w)$, meaning $x_i \leftarrow x_i + w$. At the end of the stream, we are asked to approximate $f(x)$ for a function $f : \mathbb{Z}^n \to \mathbb{R}$. In our problem, $f(x)$ is the residual $\ell_p$ norm $\|x - x_k\|_p$. Our goal is to estimate this value within a $(1 \pm \varepsilon)$ factor.

**$\ell_p$ Sparse Recovery.** Rather than estimating the value of $\|x - x_k\|_p$, here our goal is to output a $k$-sparse vector $\hat{x} \in \mathbb{R}^n$ such that $\|x - \hat{x}\|_p \leq (1 \pm \varepsilon)\|x - x_k\|_p$.

**$\ell_p$ Norm Estimation.** In the $\ell_p$ norm estimation problem, the goal is to approximate $\|x\|_p$ within a $(1 \pm \varepsilon)$ factor. This problem is well-understood and has space complexity $\Theta(n^{1-2/p}/\operatorname{poly}(\varepsilon))$, see, e.g., (Andoni et al., 2011; Ganguly & Woodruff, 2018). The algorithms for this problem will be used in our algorithm as a subroutine.

Next, we review the Count-Sketch algorithm for frequency estimation.

**Count-Sketch.** We have $k$ distinct hash functions $h_i : [n] \to [B]$ and an array $C$ of size $k \times B$. Additionally, we have $k$ sign functions $g_i : [n] \to \{-1, 1\}$. The algorithm maintains $C$ such that $C[\ell, b] = \sum_{j:h_\ell(j)=b} x_j$. The frequency estimation $\hat{x}_i$ of $x_i$ is defined to be the median of $\{g_\ell(i) \cdot C[\ell, h_\ell(i)]\}_{\ell \leq k}$.

Finally, we mention some classical results regarding singular values.

**Lemma 2.1** (Extreme singular values of Gaussian random matrices (Vershynin, 2018, Theorem 4.6.1)). *Let $G$ be an $m \times n$ ($m \geq n$) Gaussian random matrix of i.i.d. $N(0, 1)$ entries. It holds with probability at least $1 - \exp(-ct^2)$ that $\sqrt{m} - C\sqrt{n} - t \leq \sigma_{\min}(G) \leq \sigma_{\max}(G) \leq \sqrt{m} + C\sqrt{n} + t$, where $C, c > 0$ are absolute constants.*

**Lemma 2.2** (Weyl's inequality (Horn & Johnson, 2012, (7.3.13))). *Let $A, B$ be $m \times n$ matrices with $m \leq n$. It holds that $\sigma_{i+j-1}(A + B) \leq \sigma_i(A) + \sigma_j(B)$ for all $1 \leq i, j, i + j - 1 \leq m$.*

## 3 LOW-RANK RESIDUAL ERROR ESTIMATION

In this section, we consider the low-rank error estimation problem where our goal is to estimate the rank-$k$ residual error $\|A - A_k\|_F$, where $A_k$ is the best rank-$k$ approximation of $A$.

**Lower Bound.** We show that for any random $S \in \mathbb{R}^{s \times n}$, $T \in \mathbb{R}^{n \times t}$, suppose that with high constant probability we can recover $\|A - A_k\|_F$ within a factor of a $(1 + \varepsilon)$. We must then have that $s = \Omega(k/\varepsilon^2)$ and $t = \Omega(k/\varepsilon^2)$. Formally, we have the following theorem.

**Theorem 3.1.** *Suppose that for random $S \in \mathbb{R}^{s \times n}$, $T \in \mathbb{R}^{n \times t}$, there exists an algorithm $\mathcal{A}$ satisfying that $\mathcal{A}(S, T, SAT) = (1 \pm \varepsilon)\|A - A_k\|_F$ for an arbitrary $A \in \mathbb{R}^{n \times n}$ with high constant probability. Then it must hold that $s, t = \Omega(k/\varepsilon^2)$.*

To achieve this, we first show an $\Omega(k^2/\varepsilon^2)$ sketching dimension lower bound for a general sketching algorithm where the sketch has the form $S \cdot \operatorname{vec}(A)$ where $S \in \mathbb{R}^{s \times n^2}$. Then we will show that this implies an $\Omega(k^2/\varepsilon^4)$ lower bound for bilinear sketches. Consider the following two matrix distributions

$$G + c\sqrt{\varepsilon}B \qquad \text{and} \qquad G + c\sqrt{\varepsilon}B + c\alpha\sqrt{\varepsilon} \cdot uv^\top \qquad (1)$$

where

1. $G \in \mathbb{R}^{k/\varepsilon^2 \times k}$ is a random Gaussian matrix of i.i.d. $N(0,1)$ entries.
2. $\alpha$ is sampled from the distribution of the $k$-th singular values of a random Gaussian matrix $H \in \mathbb{R}^{k/\varepsilon^2 \times k}$, $u, v$ are uniformly random unit vectors and $c$ is a sufficiently large constant.
3. $B = \sum_{i=1}^{k-1} \alpha_i u_i v_i^\top$ where $\alpha_i, u_i, v_i$ are sampled from the distribution of the $i$-th singular values and singular vectors of $H \in \mathbb{R}^{k/\varepsilon^2 \times k}$ conditioned on its $k$-th singular value and singular vectors being $\alpha, u, v$.

We will first show that any linear sketching algorithm with dimension smaller than $O(k^2/\varepsilon^2)$ cannot distinguish the two distributions with high constant probability. To do so, notice that by Yao's minimax principle, we can fix the rows of our sketching matrix, and show that the resulting distributions of the sketch have small total variation distance. By standard properties of the variation distance, this implies that no estimation procedure $\mathcal{A}$ can be used to distinguish the two distributions with sufficiently large probability. Let $\mathcal{L}_1$ and $\mathcal{L}_2$ be the corresponding distribution of the linear sketch on $\mathcal{D}_1$ and $\mathcal{D}_2$ in (1), respectively. Formally, we have

**Lemma 3.2.** *Suppose that the sketching dimension of $\mathcal{L}_1$ and $\mathcal{L}_2$ is smaller than $c_1 k^2/\varepsilon^2$ for some constant $c_1$. Then we have that $d_{TV}(\mathcal{L}_1, \mathcal{L}_2) \leq 1/8$.*

*Proof.* We first consider the following two distributions.

$$ G \quad \text{and} \quad G + \beta\sqrt{\frac{\varepsilon}{k}} w z^\top $$

where $G \in \mathbb{R}^{k/\varepsilon^2 \times k}$ is a Gaussian random matrix and $w \in \mathbb{R}^{k/\varepsilon^2}$, $z \in \mathbb{R}^k$ are random Gaussian vectors, and $\beta$ is a sufficiently large constant. From Theorem 4 of (Li & Woodruff, 2016), we have that if the sketching dimension is smaller than $c_1 k^2/\varepsilon^2$, then

$$ d_{TV}(S(G), S(G + \beta\sqrt{\frac{\varepsilon}{k}} w z^\top)) \leq 1/10 $$

where $S(M)$ is the sketch on $M$. Let $S_{\text{good}} \subseteq \mathbb{R}^{k/\varepsilon^2 \times k}$ be the subset of matrix $B$ where

$$ \Pr[\alpha = \Theta(\sqrt{k}/\varepsilon) \mid B] \geq 0.99. $$

We claim that $\Pr[B \in S_{\text{good}}] \geq 0.99$. Otherwise, $\Pr[\alpha \notin \Theta(\sqrt{k}/\varepsilon)] \geq \Pr[\alpha \notin \Theta(\sqrt{k}/\varepsilon) \mid B \notin S_{\text{good}}] \Pr[B \notin S_{\text{good}}] \geq 0.01 \cdot 0.01$, contradicting the fact following from Lemma 2.1 that $\Pr[\alpha \notin \Theta(\sqrt{k}/\varepsilon)] \leq \exp(-\Omega(k/\varepsilon^2))$.

Now, fix a $B \in S_{\text{good}}$. Recall that $\alpha$ is dependent with $B$ while $u, v$ is independent with $B$, and combined with the concentration property of the $\ell_2$ norm of a Gaussian vector, we get that with probability at least 0.98, we have that $\beta\sqrt{\frac{\varepsilon}{k}} \|w\|_2 \|z\|_2 = \Theta(\alpha\sqrt{\varepsilon})$, which implies

$$ d_{TV}(S(G + c\sqrt{\varepsilon}B), S(G + c\sqrt{\varepsilon}B + c\alpha\sqrt{\varepsilon} \cdot uv^\top)) \leq 1/9 $$

for such fixed $B$. Since $\Pr[B \in S_{\text{good}}] \geq 0.99$, from the definition of total variation distance we have that $d_{TV}(\mathcal{L}_1, \mathcal{L}_2) \leq 1/9 + 0.01 < 1/8$. $\qquad\square$

We next show that if we have an algorithm $\mathcal{A}$ that computes a $(1 \pm \varepsilon)$-approximation to the rank-$(k-1)$ residual error, we then can distinguish the two distributions $\mathcal{D}_1$ and $\mathcal{D}_2$ in (1) with high probability.

**Theorem 3.3.** *Suppose that for random matrix $S \in \mathbb{R}^{s \times n^2}$, there exists an algorithm $\mathcal{A}$ satisfying that $\mathcal{A}(S, S \cdot \text{vec}(A)) = (1 \pm \varepsilon)\|A - A_k\|_F$ for an arbitrary $A \in \mathbb{R}^{n \times n}$ with high constant probability. Then it mush hold that $s = \Omega(k^2/\varepsilon^2)$.*

*Proof.* By Yao's minimax principle, we may assume that $A$ is drawn from the distributions in (1) and $S$ is a deterministic sketching matrix. Note that for both matrix families, the drawn matrix $A$ is rank-$k$ with high probability, and hence the rank-$(k-1)$ residual error of $A$ is equal to its smallest singular value. We shall examine the smallest singular value for the two distributions. If $A \sim \mathcal{D}_1$, from Lemma 2.1 and Weyl's inequality (Lemma 2.2), we have that with high probability

$$ \sigma_{\min}(G + c\sqrt{\varepsilon}B) \leq \sigma_{\max}(G) + \sigma_{\min}(c\sqrt{\varepsilon}B) = \sigma_{\max}(G) \leq \frac{\sqrt{k}}{\varepsilon} + C_1\sqrt{k} $$

since $B$ is rank $k-1$. On the other hand, when $A \sim \mathcal{D}_2$, from the definition of $B, \alpha, u, v$, we see that $B + \alpha uv^\top$ is also a Gaussian matrix and thus $G + c\sqrt{\varepsilon}(B + \alpha uv^\top)$ is a Gaussian random matrix with entries $N(0, 1 + c^2\varepsilon)$. Again, from Lemma 2.1, we have with high probability that

$$\sigma_{\min}(G + c\sqrt{\varepsilon}B + c\alpha\sqrt{\varepsilon} \cdot uv^\top) \geq \sqrt{1 + c^2\varepsilon} \cdot (\frac{\sqrt{k}}{\varepsilon} - C\sqrt{k}) \geq \frac{\sqrt{k}}{\varepsilon} + C_2\sqrt{k},$$

provided that $c^2/2 \geq C_2 + C\sqrt{1 + c^2\varepsilon}$. This is satisfied by choosing $c$ to be large enough. It follows immediately that $\mathcal{A}$ can distinguish the two distributions $\mathcal{D}_1$ and $\mathcal{D}_2$ and the theorem then follows from Lemma 3.2. □

*Proof of Theorem 3.1.* Suppose that $M$ is drawn from the distribution in (1). By Yao's minimax principle, we can assume that $S$ and $T$ are deterministic sketching matrices. Let $A \in \mathbb{R}^{k/\varepsilon^2 \times k/\varepsilon^2}$ be one of the following two matrices with the same probability

$$[M \quad \mathbf{0}] \qquad \text{or} \qquad \begin{bmatrix} M^\top \\ \mathbf{0} \end{bmatrix}.$$

We first consider the case when $A = [M \quad \mathbf{0}]$, then $SAT = SMT'$, where $T'$ is the first $k$ rows of $T$. Let $T''$ be a submatrix of $T'$ consisting of a maximal linearly independent subset of columns of $T'$. Then $T''$ has $t' \leq k$ columns. Furthermore, since the columns of $T'$ are linear combinations of the columns of $T''$, we can recover $SMT'$ from $SMT''$. Hence, $\mathcal{A}(S, T, SAT)$ induces an algorithm $\mathcal{A}'(S, T'', SMT'')$ of the same output, which estimates $\|A - A_k\|_F = \|M - M_k\|_F$ up to a $(1 \pm \varepsilon)$-factor. Next, note that the $(i, j)$-th entry of $SMT''$ is equal to $\langle S_i T''^\top_j, \text{vec}(M) \rangle$. Therefore, from Theorem 3.3, we have that $s \cdot t' = \Omega(k^2/\varepsilon^2)$. Since $t' \leq k$, it follows immediately that $s = \Omega(k/\varepsilon^2)$.

A similar argument for $A = [\begin{smallmatrix} M^\top \\ \mathbf{0} \end{smallmatrix}]$ yields that $t = \Omega(k/\varepsilon^2)$. □

**Upper Bound.** We shall give an $O(k^2/\varepsilon^4)$ upper bound for bilinear sketches. We first recall the definition of Projection-Cost Preserving sketches (PCPs).

**Definition 3.4** ((Cohen et al., 2015)). *Given a matrix $A \in \mathbb{R}^{n \times d}$, $\varepsilon > 0$, $c \geq 0$ and an integer $k \in [d]$, a sketch $S \in \mathbb{R}^{s \times n}$ is an $(\varepsilon, c, k)$-column projection-cost preserving sketch of $A$ if for all rank-$k$ projection matrices $P$, $(1 - \varepsilon)\|A(I - P)\|_F^2 \leq \|SA(I - P)\|_F^2 + c \leq (1 + \varepsilon)\|A(I - P)\|_F^2$.*

**Lemma 3.5** ((Cohen et al., 2015; Musco & Musco, 2020)). *Let $S \in \mathbb{R}^{m \times n}$ be drawn from any of the following matrix families. Then with probability $1 - \delta$, $S$ is an $(\varepsilon, 0, k)$-projection-cost-preserving sketch of $A$.*

1. *$S$ is a dense Johnson-Lindenstrauss (JL) matrix, with $c = 0$, $m = O((k + \log(1/\delta))/\varepsilon^2)$ and each element is chosen independently and uniformly in $\pm\sqrt{1/m}$.*
2. *$S$ is a COUNTSKETCH with $c = 0$, $m = O(k^2/(\varepsilon^2\delta))$, where each column has a single $\pm 1$ in a random position.*
3. *$S$ is an ONSAP sparse subspace embedding matrix (Nelson & Nguyen, 2013), with $c = 0$ and $m = O(k \log(k/\delta)/\varepsilon^2)$, where each column has $s = O(\log(1/\varepsilon))$ random $\pm 1/\sqrt{s}$.*

We remark that it is easy to see from the definition that if $S_1$ and $S_2$ are both $(\varepsilon, 0, k)$-PCP sketches of $A$, then $S_1 S_2$ is an $(O(\varepsilon), 0, k)$-PCP sketch of $A$.

Our algorithm is as follows. Suppose that $S$ and $T$ are matrices such that $S$ and $T^\top$ both satisfy the condition in Definition 3.5. We then compute the rank-$k$ error $\|SAT - [SAT]_k\|_F$ and use it as our final estimate. To show the correctness of our algorithm, we first prove the following lemma.

**Lemma 3.6.** *Suppose that $S$ is an $(\varepsilon, c, k)$ projection-cost preserving sketch of $SA$. Then we have*

$$\|A - A_k\|_F^2 \leq \|SA - [SA]_k\|_F^2 + c \leq (1 + \varepsilon)\|A - A_k\|_F^2$$

*Proof.* Suppose that $\widetilde{P}$ is the minimizer of $\|SA(I - P)\|_F^2$ over all rank-$k$ projections $P$, and $P^*$ is the minimizer of $\|A(I - P)\|_F^2$. Since $S$ is an $(\varepsilon, c, k)$ projection cost preserving sketch of $A$, we have that

$$\|A(I - P^*)\|_F^2 \leq \|A(I - \widetilde{P})\|_F^2 \leq \|SA(I - \widetilde{P})\|_F^2 + c$$
$$\leq \|SA(I - P^*)\|_F^2 + c \leq (1 + \varepsilon)\|A(I - P^*)\|_F^2.$$

---

**Algorithm 1:** $(1 \pm \varepsilon)$-approximator for $\|x_{-k}\|_p^p$

---

1   Set $b = \Theta(\varepsilon^{-2p/(p-1)} k^{2/p} n^{1-2/p})$ and $\ell = O(\log n)$;
2   Initialize $\ell \cdot b$ buckets $z_{1,1}, \ldots, z_{b,\log n}$ to 0;
3   Initialize $\ell$ pairwise independent hash functions $h_i : [n] \to [b]$;
4   Initialize $\ell$ 4-wise hash functions $s_j : [n] \to \{-1, 1\}$;
5   Initialize a $(1 \pm \varepsilon)$-$\ell_p$ estimation algorithm $\mathcal{A}$ with $\widetilde{O}(n^{1-2/p}/\operatorname{poly}(\varepsilon))$ space [e.g., (Andoni et al., 2011)];
6   **foreach** $(i, v)$ *update comes* **do**
7      **for** $j \leftarrow 1$ **to** $\ell$ **do**
8          $z_{h_j(i),j} \leftarrow z_{h_j(i),j} + v \cdot s_j(i)$;
9      **end**
10      Perform the update $(i, v)$ in $\mathcal{A}$;
11   **end**
12   **for** $i \leftarrow 1$ **to** $\ell$ **do**
13      $\hat{x}_i = \operatorname{median}_j \{s_j(i) \cdot z_{h_j(i),j}\}$
14   **end**
15   Choose the top $k$ coordinates of $\hat{x}$ to form set $J$ ;
16   **foreach** $j \in J$ **do**
17      Perform the update $(j, -\hat{x}_j)$ in $\mathcal{A}$.
18   **end**
19   **return** *Output of $\mathcal{A}$*;

---

Next, from the definitions of $\widetilde{P}$ and $P^*$, we have that $\|SA - [SA]_k\|_F^2 = \|SA(I - \widetilde{P})\|_F^2$ and $\|A - A_k\|_F^2 = \|A(I - P^*)\|_F^2$. The desired result follows.     $\square$

**Theorem 3.7.** *There exist random matrices $S \in \mathbb{R}^{O(k/\varepsilon^2) \times n}$ and $T \in \mathbb{R}^{d \times O(k/\varepsilon^2)}$ such that with high constant probability*

$$\|A - A_k\|_F^2 \leq \|SAT - [SAT]_k\|_F^2 \leq (1 + \varepsilon)\|A - A_k\|_F^2 . \tag{2}$$

*Moreover, the sketch $SAT$ can be computed in $\operatorname{nnz}(A) + \operatorname{poly}(k/\varepsilon)$ time.*

*Proof.* We construct $S = S_1 S_2$, where $S_1 \in \mathbb{R}^{O(k/\varepsilon^2) \times O(k^2/\varepsilon^2)}$ is a JL matrix and $S_2 \in \mathbb{R}^{O(k^2/\varepsilon^2) \times n}$ is the COUNT-SKETCH matrix in Lemma 3.5. Then $T^\top = T_1^\top T_2^\top$ is the same construction with $S$ but replacing $n$ with $d$. Because $S_2$ and $T_2$ are both COUNT-SKETCH matrices, we can compute $S_2 A T_2$ in $\operatorname{nnz}(A)$ time. Then note that $S_2 A T_2$ are matrices with size $O(k^2/\varepsilon^2) \times O(k^2/\varepsilon^2)$, so we can compute $SAT = S_1 S_2 A T_2 T_1$ in $\operatorname{nnz}(A) + \operatorname{poly}(k/\varepsilon)$ time.

We next consider the accuracy. From Lemma 3.5 we have that with high constant probability $S, T$ are both rank-$k$ PCPs with error $\varepsilon$. Hence we first have

$$\|A - A_k\|_F^2 \leq \|SA - [SA]_k\|_F^2 \leq (1 + \varepsilon)\|A - A_k\|_F^2 .$$

Applying Lemma 3.6 to $T^\top$ and $A^\top S^\top$ yields that

$$\|SA - [SA]_k\|_F^2 \leq \|SAT - [SAT]_k\|_F^2 \leq (1 + \varepsilon)\|SA - [SA]_k\|_F^2 .$$

Combining the two equations above leads to

$$\|A - A_k\|_F^2 \leq \|SAT - [SAT]_k\|_F^2 \leq (1 + O(\varepsilon))\|A - A_k\|_F^2$$

and the claimed result then follows immediately by rescaling $\varepsilon$.     $\square$

## 4   $F_p$ RESIDUAL ERROR ESTIMATION

We next consider the $F_p$ residual error estimation task where our goal is to estimate the $k$-residual error $\|x_{-k}\|_p$ up to a $(1 \pm \varepsilon)$-factor.

The algorithm is shown in Algorithm 1. At a high level, we use the classical COUNTSKETCH to estimate the frequency of each coordinate of $x$. Then we select the top $k$ coordinates that have the $k$

largest estimated values. In parallel we run an $F_p$ estimation algorithm $\mathcal{A}$ and then subtract these coordinates with the estimated value. We then use the output of $\mathcal{A}$ to be our final estimation of $\|x_{-k}\|_p^p$. Suppose that $I$ is the set of coordinates that are the genuine $k$ largest coordinates, and $J$ is the set of candidates we choose. In the remainder of this section, we shall show that $\|x - x_J\|_p$ is within a $(1 \pm \varepsilon)$ factor of $\|x - x_I\|_p = \|x_{-k}\|_p$. Hence, if we run an $F_p$ frequency estimation algorithm in parallel (see, e.g., (Ganguly & Woodruff, 2018)) and after subtracting the estimated frequency $\hat{x}_j$ on each of the coordinates in $J$, we will obtain the $k$ residual error up to a $(1 \pm \varepsilon)$ factor.

For the purpose of exposition, we first assume $b = \Theta(\varepsilon^{-2}k^{2/p}n^{1-2/p})$ and will show that it gives a $O(1 + \varepsilon^{1-1/p})$-approximation. Then, after normalization, we get the desired bound. We first bound the error of COUNTSKETCH for each coordinate. The following lemma is a standard fact of COUNTSKETCH. The corollary below easily follows from Hölder's inequality.

**Lemma 4.1** ((Minton & Price, 2014)). *Suppose that $\ell \geq c \log n$ for a constant $c$. Then with probability at least $9/10$, we have that $|\hat{x}_i - x_i| \leq \|x_{-k}\|_2/\sqrt{b}$ holds for all $i$ simultaneously.*

**Corollary 4.2.** *If we set the number of buckets $b = \Theta(\varepsilon^{-2}k^{2/p}n^{1-2/p})$, then with probability at least $9/10$, we have that $|\hat{x}_i - x_i| \leq \varepsilon\|x_{-k}\|_p/(c_1 k^{1/p})$, where $c_1$ is a universal constant.*

For an index set $T \subseteq [n]$, we define $S_T = \sum_{t \in T} |x_t|^p$. From the definition of the set $I$ and $J$, we have that $S_J \leq S_I$. For the other direction, we have

**Lemma 4.3.** *It holds that $S_J \geq S_I - O(\varepsilon^{1-1/p}) \cdot \|x_{-k}\|_p^p$.*

*Proof.* It follows from Lemma 4.2 and the definitions of $I$ and $J$ that if some $i \in I$ is replaced by some $j \in J$, we must have (i) $|x_i - x_j| \leq \varepsilon\frac{\|x_{-k}\|_p}{c_1 k^{1/p}}$, and (ii) $|x_i|^p \geq |x_j|^p \geq \left| |x_i| - 2\varepsilon\frac{\|x_{-k}\|_p}{c_1 k^{1/p}} \right|^p$.

We claim that $|x_i|^p \leq 2\|x_{-k}\|_p^p$ for all $i \in I \setminus J$. If not, suppose that $|x_i| > 2^{1/p}\|x_{-k}\|_p$ for some $i \in I \setminus J$, we would obtain an estimate $\hat{x}_i$ with $|\hat{x}_i| > (2^{1/p} - \varepsilon/(c_1 k^{1/p}))\|x_{-k}\|_p$ by Corollary 4.2. Since $i \notin J$, the estimates of $x_j$ for every $j \in J$ must be at least $(2^{1/p} - \varepsilon/(c_1 k^{1/p}))\|x_{-k}\|_p$, which further implies that there exists some $j \notin I$ such that $|x_j| \geq (2^{1/p} - 2\varepsilon/(c_1 k^{1/p}))\|x_{-k}\|_p$. This contradicts the fact that $\|x_{-k}\|_p \geq |x_j|$. Hence it must hold that $|x_i|^p \leq 2\|x_{-k}\|_p^p$ for all $i \in I \setminus J$.

We next decompose $I$ as $I = (I \cap J) \cup T_0 \cup T_1 \cup \cdots \cup T_m \cup T_{m+1}$, where $m = O(\log(k/\varepsilon))$ is such that $2^{m-1} \leq k/(10\varepsilon) < 2^m$,

$$T_\ell = \left\{ i \in I \setminus J : \frac{\|x_{-k}\|_p^p}{2^\ell} < |x_i|^p \leq \frac{\|x_{-k}\|_p^p}{2^{\ell-1}} \right\}, \quad \ell = 0, 1, 2, \ldots, m$$

and

$$T_{m+1} = \left\{ i \in I \setminus J : |x_i|^p \leq \frac{10\varepsilon\|x_{-k}\|_p^p}{k} \right\}.$$

For $T_{m+1}$ we have that

$$\sum_{i \in T_{m+1}} |x_i|^p \leq k \cdot \frac{10\varepsilon\|x_{-k}\|_p^p}{k} = 10\varepsilon\|x_{-k}\|_p^p.$$

Next consider $T_\ell$. Note that

$$\sum_{\substack{i \in T_\ell \\ i \text{ is displaced by } j}} |x_j|^p \leq \|x_{-k}\|_p^p,$$

and it must hold that $|T_\ell| = O(2^\ell)$. Suppose that $i \in T_\ell$ is displaced by $j \in J$. It then follows from the above discussion that

$$|x_i|^p - |x_j|^p \leq |x_i|^p - \left| |x_i| - 2\varepsilon\frac{\|x - x_k\|_p}{c_1 k^{1/p}} \right|^p \leq |x_i|^{p-1} \cdot 2p\varepsilon\frac{\|x - x_k\|_p}{c_1 k^{1/p}}$$

$$\leq \frac{\|x - x_k\|_p^{p-1}}{2^{(\ell-1)\cdot\frac{p-1}{p}}} \cdot 2p\varepsilon\frac{\|x - x_k\|_p}{c_1 k^{1/p}} = 2p\varepsilon\frac{\|x_{-k}\|_p^p}{c_1 (2^{1-\frac{1}{p}})^{\ell-1} k^{\frac{1}{p}}}$$

Taking the sum we get that $\sum_{i \in T_\ell} |x_i|^p - \sum_{\substack{i \in T_\ell \\ i \text{ is displaced by } j}} |x_j|^p \leq O(2^\ell) \cdot 2p\varepsilon \frac{\|x_{-k}\|_p^p}{c_1(2^{1-\frac{1}{p}})^{\ell-1}k^{\frac{1}{p}}} = O\left(\frac{\varepsilon 2^{\ell/p}\|x_{-k}\|_p^p}{k^{1/p}}\right)$. Summing over $\ell$ yields that $S_I - S_J \leq O(\varepsilon^{1-1/p}) \cdot \|x_{-k}\|_p^p$, which is what we need. $\qquad\square$

Suppose that we have $S_I - S_J \leq O(\varepsilon) \cdot \|x_{-k}\|_p^p$ (we normalize $\varepsilon$ here). Then this means that $\left| \|x - x_J\|_p^p - \|x_{-k}\|_p^p \right| = \left| \|x - x_J\|_p^p - \|x - x_I\|_p^p \right| \leq \varepsilon \|x_{-k}\|_p^p$. Then, from the discussion above we also have that for each $j \in J$, $|\hat{x}_j - x_j| \leq \varepsilon^{p/(p-1)} \frac{\|x_{-k}\|_p}{c_1 k^{1/p}}$, which means that $\left| \|x - \hat{x}_J\|_p^p - \|x - x_J\|_p^p \right| \leq k \cdot \frac{\varepsilon^{p/(p-1)}\|x_{-k}\|_p^p}{c_1^p k} = O(\varepsilon^{p/(p-1)} \|x_{-k}\|_p^p)$. Therefore, $\left| \|x - \hat{x}_J\|_p^p - \|x_{-k}\|_p^p \right| = O\left(\varepsilon \|x_{-k}\|_p^p\right)$, whence we see that any $(1 \pm \varepsilon)$-approximation of $\|x - \hat{x}_J\|_p^p$ is an $(1 \pm O(\varepsilon))$-approximation of $\|x_{-k}\|_p^p$. We can now conclude with the following theorem.

**Theorem 4.4.** *There is an algorithm that uses space $\widetilde{O}(\varepsilon^{-2p/(p-1)}k^{2/p}n^{1-2/p})$ and outputs a $(1 \pm \varepsilon)$-approximation of the $k$-residual error $\|x_{-k}\|_p^p$ with high constant probability.*

$\ell_p$ **Sparse Recovery.** Recall that in our algorithm, the vector $\hat{x}_J$ is $k$-sparse and satisfies that $\|x - \hat{x}_J\| \leq (1+\varepsilon)\|x_{-k}\|_p^p$, which means that Algorithm 1 actually solves the $\ell_p$ sparse recovery problem. We have the following theorem.

**Theorem 4.5.** *There is an algorithm that uses space $\widetilde{O}(\varepsilon^{-2p/(p-1)}k^{2/p}n^{1-2/p})$ and solves the $(1+\varepsilon)$-$\ell_p$ sparse recovery problem with high constant probability.*

Below we show an $\Omega(k^{2/p}n^{1-2/p})$ lower bound for the $\ell_p$ sparse recovery problem with a constant approximation factor. To achieve this, we consider the Gap-infinity problem in (Bar-Yossef et al., 2004).

**Definition 4.6** (Gap-infinity problem, (Bar-Yossef et al., 2004))**.** *There are two parties, Alice and Bob, holding vectors $a, b \in \mathbb{Z}^n$ respectively, and their goal is to decide if $\|a - b\|_\infty \leq 1$ or $\|a - b\|_\infty \geq s$.*

**Theorem 4.7** ((Bar-Yossef et al., 2004))**.** *Any protocol that solves the Gap-infinity problem with probability at least $9/10$ must have $\Omega(n/s^2)$ bits of communication.*

The lower bound also holds if we assume there is exactly 1 or 0 coordinates $i$ satisfying $|a_i - b_i| \geq s$, each case occurring with constant probability. The work of (Bar-Yossef et al., 2004) provides an information cost lower bound for this problem, which, together with the direct sum theorem ((Chakrabarti et al., 2001; Bar-Yossef et al., 2004)), leads to the following corollary.

**Corollary 4.8.** *Suppose that there are $t$ independent instances of the Gap-infinity problem. If Alice and Bob can solve a constant fraction of the instances with probability at least $9/10$, then they must have $\Omega(tn/s^2)$ bits of communication.*

Now we are ready to prove a bit lower bound for the $\ell_p$-sparse recovery problem.

**Theorem 4.9.** *Suppose that $c$ is a sufficiently small constant. Any algorithm that solves the $(1+c)$ $\ell_p$ sparse recovery problem with high constant probability requires $\Omega(k^{2/p}n^{1-2/p})$ bits of space.*

*Proof.* We reduce the multi-instance Gap-infinity problem to the $\ell_p$ sparse recovery problem. Suppose that there are $k$ instances of the Gap-infinity problem $(a^i, b^i)$ with length $n/k$ and $s = (n/k)^{1/p}$. Let $z^i = a^i - b^i$ ($i \in [n/k]$) and $x \in \mathbb{Z}^n$ be the resulting vector after concatenating all $z^i$. Suppose that $\hat{x}$ is a $k$-sparse vector which satisfies $\|x - \hat{x}\|_p^p \leq (1+c)\|x - x_k\|_p^p$, where $c$ is a sufficiently small constant. We shall show how we can solve, using $\hat{x}$, a constant fraction of copies of the Gap-infinity problem on $(a^i, b^i)$. As mentioned above, we can assume for each instance $(a^i, b^i)$ that there is exactly 1 or no coordinates $j$ satisfying $|(a^i)_j - (b^i)_j| \geq s$, each case occurring with constant probability. Under this assumption, with high constant probability, there are $\Theta(k)$ instances for which $\|a^i - b^i\|_\infty = s$ and $\|x_{-k}\|_p^p = O(n)$. Note that for a coordinate $x_i$, if $|x_i - \hat{x}_i| \geq s/2$, it will contribute $\Omega(n/k)$ to $\|x - \hat{x}\|_p^p$. Thus, for a $(1+c)$-approximation, this event can only happen at most $O(k)$ times, which means that from the solution $\hat{x}$, we can solve a constant fraction of the instances $(a^i, b^i)$. The theorem then follows from Corollary 4.8. $\qquad\square$

Table 1: Performance of our algorithm and (Andoni & Nguyen, 2013) on MovieLens 100K data and KOS data, respectively.

| | $k = 5$ | $k = 10$ | $k = 20$ | | $k = 5$ | $k = 10$ | $k = 20$ |
|---|---|---|---|---|---|---|---|
| $\varepsilon$(Ours, $m = 50$) | 0.146 | 0.295 | 0.545 | $\varepsilon$(Ours, $m = 50$) | 0.135 | 0.287 | 0.543 |
| $\varepsilon$(Ours, $m = 100$) | 0.074 | 0.149 | 0.292 | $\varepsilon$(Ours, $m = 100$) | 0.068 | 0.140 | 0.279 |
| $\varepsilon$(AN13, $m = 50$) | 0.135 | 0.287 | 0.541 | $\varepsilon$(AN13, $m = 50$) | 0.141 | 0.288 | 0.540 |
| $\varepsilon$(AN13, $m = 100$) | 0.070 | 0.149 | 0.288 | $\varepsilon$(AN13, $m = 100$) | 0.067 | 0.138 | 0.286 |
| Runtime (Ours, $m = 50$) | 0.105s | 0.102s | 0.105s | Runtime (Ours, $m = 50$) | 0.117s | 0.114s | 0.120s |
| Runtime (Ours, $m = 100$) | 0.106s | 0.105s | 0.109s | Runtime (Ours, $m = 100$) | 0.126s | 0.130s | 0.123s |
| Runtime (AN13, $m = 50$) | 0.377s | 0.381s | 0.388s | Runtime (AN13, $m = 50$) | 0.399s | 0.414s | 0.397s |
| Runtime (AN13, $m = 100$) | 0.735s | 0.735s | 0.728s | Runtime (AN13, $m = 100$) | 0.747s | 0.744s | 0.744s |
| Streaming LRA ($m = 50$) | 0.141s | 0.149s | 0.151s | Streaming LRA ($m = 50$) | 0.651s | 0.657s | 0.660s |
| Randomized SVD | 0.056s | 0.061s | 0.075s | Randomized SVD | 0.199s | 0.218s | 0.230s |
| SVD | 1.180s | | | Runtime of SVD | 22.070s | | |

We have shown a lower bound in terms of total bits of space. We next show that such a lower bound can be converted to a sketching dimension lower bound, for which we need the following lemma.

**Lemma 4.10** ((Price & Woodruff, 2011, Lemma 5.2)). *A lower bound of $\Omega(b)$ bits for the sparse recovery bit scheme implies a lower bound of $\Omega(b/\log n)$ for regular sparse recovery with failure probability $\delta - 1/n$.* [1]

Our theorem follows immediately.

**Theorem 4.11.** *Suppose that $c$ is a sufficiently small constant. Any algorithm that solves the $(1 + c)$ $\ell_p$ sparse recovery problem with high constant probability requires $\widetilde{\Omega}(k^{2/p}n^{1-2/p})$ measurements.*

## 5 EXPERIMENTS

In this section, we consider experiments for the low-rank residual error estimation problem on real-world datasets. All of our experiments were done in Python and conducted on a device with a 3.30GHz CPU and 16GB RAM. We will use the following dataset.

- **KOS data.**[2] A word frequency dataset. The matrix represents word frequencies in blogs and has dimensions $3430 \times 6906$ with 353160 non-zero entries.
- **MovieLens 100K.** (Harper & Konstan, 2016) A movie ratings dataset, which consists of a preference matrix with 100,000 ratings from 611 users across 9,724 movies.

As discussed in the previous section, our algorithm follows the same framework as that in (Andoni & Nguyen, 2013): compute the sketch $SAT$ and then compute the rank-$k$ residual error on $SAT$: $\|SAT - [SAT]_k\|_F$. However, the work of Andoni & Nguyen (2013) only gives an $O(k^2/\varepsilon^6)$ upper bound for this bilinear sketch, while we have shown any sketch matrices with the rank-$k$ projection-cost preserving property suffice, which allows for an $O(k^2/\varepsilon^4)$ upper bound and the use of extremely sparse sketching matrices with this size.

The result is shown in Table 1. We define the error $\varepsilon = $ (Output of the Algorithm)$/\|A - A_k\|_F - 1$, and take an average over 10 independent trials. Since the regime of interest is $k \ll n, d$, we vary $k$ among $\{5, 10, 20\}$ and set the sketching size to be $m = 50, 100$. For the work of Andoni & Nguyen (2013), we set the matrices $S, T$ to be random Gaussian matrices and for ours we set the $S, T$ to be the OSNAP matrices (Nelson & Nguyen, 2013; Cohen, 2016) with $s = 2$. The result shows that while the error of the two ways is almost the same, the runtime of ours is about 4- to 7-fold faster than the algorithm in (Andoni & Nguyen, 2013). This is because, in our algorithm, the sketching matrix is extremely sparse, where each column has only $O(1)$ non-zero entries.

---

[1]The theory in (Price & Woodruff, 2011) is for $p \leq 2$, however, the argument still goes through for constant $p > 2$ unchanged.

[2]The Bag of Words Data Set from the UCI Machine Learning Repository.

ACKNOWLEDGMENT

Y. Li is supported in part by the Singapore Ministry of Education (AcRF) Tier 2 grant MOE-T2EP20122-0001 and Tier 1 grant RG75/21. H. Lin and D. Woodruff would like to thank support from the National Institute of Health (NIH) grant 5R01 HG 10798-2. D.W. also did part of this work while visiting the Simons Institute for the Theory of Computing.

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
