# OpenReview forum: "Optimal Sketching for Residual Error Estimation for Matrix and Vector Norms"
_ICLR.cc/2024/Conference — ICLR 2024 poster_

### Official Review · Reviewer_k2z9 · 2023-10-22

**Soundness:** 3 good
**Presentation:** 3 good
**Contribution:** 2 fair
**Rating:** 5
**Confidence:** 4

**Summary:**

This paper considers the problem of estimating the Frobenius norm residual error of the best rank-$k$ low-rank matrix approximation and the $\ell_p$-norm residual error of the best $k$-sparse vector approximation. The main result, for the matrix case, is that best rank-$k$ approximation error can be approximated to relative error $\varepsilon$ using a bilinear compression $SAT \in \mathbb{R}^{O(k/\varepsilon^2) \times O(k/\varepsilon^2)}$ of $A$ by two projection-cost-preserving sketches $S$ and $T$. The new result improves the dependence on $\varepsilon$ from (Andoni and Nguyen, 2013); a new lower bound establishes the optimality of the current method.

**Strengths:**

This is a nice work of theoretical computer science. The writing is good and reasonably clear. The fact that the projection-cost-preserving sketch technology allows for an improvement of the Andoni–Nguyen result is very interesting and, to the best of my knowledge, novel. The lower bound is nice as well, and it establishes the optimality of the proposed approach (though I have my concerns about the correctness of Lemma 3.2).

**Weaknesses:**

As a work of _theoretical computer science_, this paper is perfectly adequate. As a work of _machine learning_, it has a few deficiencies.

### Use Case

When is the proposed method useful? The story the authors tell feels a bit farfetched to me:

- Using a generalized Nyström approximation and OSNAP sketches, a rank-$k$ approximation can be computed in a single pass over the matrix. For a dense matrix stored on disk, the time cost is roughly $O(mn \log k + (m+n) k^2)$; the cost of the proposed residual estimate is roughly $O(mn)$. The difference between these costs is not very significant, particularly if $k \ll m,n$.
- The authors suggest data streams as a potential application. Indeed, the proposed approach does give a way of estimating the residual error from a stream in low space. After which, the authors say one can use the residual estimate to decide whether or not low-rank approximation is worth it. But this raises two objections: 1) If one is going to compute a low-rank approximation anyways, then one will have to eat the space cost of storing such an approximation eventually. The proposed low-space diagnostic doesn't seem useful if, conditional on a positive outcome, actually computing the approximation needs more space. 2) Often in the streaming setting, we assume we get only a single pass over the data. The authors approach requires two passes: one to compute the diagnostic and a second to compute a low-rank approximation.

It may be possible that there is some use case where the proposed approach is natural, but it seems to me that the proposed method probably has fairly limited usefulness in practice.

### CountSketch

I find the author's use of CountSketch to be questionable, particularly in the numerical experiments. In my experience, an OSNAP (even with just 2 entries of sparsity per column) is dramatically more safe and effective than CountSketch in practice.

To demonstrate this, I ran my own experiment. Let $A = \operatorname{diag}( i^{-6} : i=1,2,\ldots,10^4)$ and, following Table 1, consider $m = 50$ and $k = 20$. With CountSketch, the estimate $\| SAT - (SAT)_k \|_F$ is a factor of **58** smaller than the true residual error $\| A - A_k \|_F$. By contrast, OSNAP with _just 2 nonzeros per column_ only underestimates the residual error by a factor of 2! For the proposed use case of screening whether or not significant compression is possible by low-rank approximation, an estimate within a factor of two is probably useful; one that is wrong by more than an order of magnitude is not.

This is a paper about theoretical computer science: Rigorous analysis of computational methods so that they work in practice even on worst-case inputs. The use of a plain CountSketch in the numerical experiments, now demonstrated to be able to severely underestimate the residual error in practice, should be removed in favor of the OSNAP or, perhaps, the composite CountSketch–Gaussian sketch.

As a side note, the introduction claims that it will prove that the method works if $S$ and $T$ are chosen to be OSNAP's. But this is omitted from the main text. This should be included, because OSNAP's are often the most effective sketching matrices in practice.

### Numerical Evaluation

The runtime of a full, dense SVD is not the appropriate comparison in Table 1. Rather, the appropriate comparison is a _randomized low-rank approximation_ such as the randomized SVD or generalized Nyström (with OSNAP's, if the latter).

### Accuracy

As I mention above and as the authors tacitly admit with their numerical experiments, an error estimate which is within a factor of two is often sufficient, particularly if the proposed use case is determined whether or not to run a low-rank approximation algorithm. Viewed in this light, the improvement of the sketch dimension from $O(k/\varepsilon^3)$ to $O(k/\varepsilon^2)$ in the present work is not that meaningful in practice.

### Sparse Approximation

The residual error for $k$-sparse approximation of vectors in $\ell_p$ norms for $p > 2$ is very much a question of interest to theoretical computer science. I fail to see any interest for practical problems in machine learning.

### Conclusion

Ultimately, I feel like this paper would be more at home at a dedicated TCS publication venue. I'm not sure when I would actually run the author's proposed method in practice, and the improvement from $O(k/\varepsilon^3)$ to $O(k/\varepsilon^2)$ has limited practical implications as well. I have significant concerns with the use of plain CountSketch in the numerical experiments. If the numerical experiments were fixed to avoid the use of plain CountSketch, I would be inclined to give a soft recommendation for acceptance.

**Questions:**

### Is Lemma 3.2 correct?

The proof of Lemma 3.2 is not convincing to me in several ways. First, why does $d_{TV}(S(G), S(G+\beta \sqrt{\varepsilon/k} \cdot wz^\top) \le 1/10$ and $\beta \sqrt{\varepsilon/k} \cdot \|w\|\|z\|$ imply $d_{TV}(S(G), S(G + c\alpha\sqrt{\varepsilon} \cdot uv^\top)) \le 1/9$. I believe the ultimate result is probably true, but I don't following the line of reasoning.

I'm not sure data processing and linearity of the sketch is enough to conclude that $d_{TV}(\mathcal{L}_1,\mathcal{L}_2)$. The perturbation $c\sqrt{\varepsilon}B$ is applied to both $G$ and $G + c\alpha \sqrt{\varepsilon} uv^\top$ and it is _dependent_ with the latter of these. I'm not sure exactly what the author's argument is. It appears that the authors are making a claim of the form:

> If $d_{\rm TV}(X,Y) \le \varepsilon$, then $d_{\rm TV}(X+Z,Y+Z)\le \varepsilon$.

This claim is certainly not true. Let $X$ be a random variable which is never $0$ and take $X = -Y = Z$. Then $d_{\rm TV}(X,Y) = 0$ but $d_{\rm TV}(X+Z,Y+Z) = d_{\rm TV}(2X,0) = 1$.

I'm not sure how exactly the authors purport to conclude the proof of lemma 3.2. The authors should revise this proof to make unassailable the correctness of this result.

---

> ### Author Response · Authors · 2023-11-17
> **Response to Reviewer k2z9**
>
> We thank the reviewer for their thoughtful and detailed comments. We have uploaded a new version of our submission, where we added OSNAP in Lemma 3.5 and replaced the CountSketch matrix with OSNAP in our experiments section.
>
> - The proof of Lemma 3.2: We thank the reviewer for pointing this out. We have added more details to the proof of this lemma. Please check our new version of the submission. At a high level, here we have that $d_{TV}(X,Y)$ is small and wish to prove that $d_{TV}(X+Z,Y+Z)$ is small, where $Z$ is independent of $X$ but not of $Y$. In fact, we can show that $d_{TV}(X+Z,Y+Z) = d_{TV}(X,Y)$ is small when conditioned on a fixed $Z$ (the distribution of $Y$ depends on the choice of $Z$) over an overwhelming number of choices of $Z$. This is enough to conclude that the unconditioned value $d_{TV}(X+Z,Y+Z)$ is small.
>
> - The experiments: the reason we use a plain CountSketch matrix is that we found it has good performance for the two real-world datasets. We agree with the reviewer's point that OSNAP has a better worst-case guarantee. Hence, we have replaced CountSketch with ONSAP in our new version of the experiments. (Note that the new experiments are conducted on a different device, so the runtime is incomparable with that of the previous version.)
>
> - On use cases: there are natural settings in which we do not necessarily need to compute a low-rank approximation of all of the input matrices $A_1, A_2, \cdots, A_m$. Instead, we only need to compute an $\alpha$-fraction of them, based on which of them has a good low-rank approximation. In the case when $\alpha$ is small (such as $0.1$) and computing an actual low-rank approximation is much more expensive than estimating its cost, our algorithm will achieve a factor of roughly $\alpha$ savings in its resource requirements. In the streaming model, the space we need for our algorithm is $\mathrm{poly}(k)$, while if we want to compute a rank-$k$ approximation in a stream, the space we need would be $\Omega((n + m) \cdot k)$, which is larger than $\mathrm{poly}(k)$ when $k \ll m, n$. Also, the overall runtime of our algorithm for our rank-$k$ residual estimation algorithm is $O(\mathrm{nnz}(A) + \mathrm{poly}(k))$, while for a rank-$k$ approximation, the overall runtime is $O(\mathrm{nnz}(A) + (n + m) \cdot \mathrm{poly}(k))$. Hence, if the matrix $A$ is sparse enough, the second term of $\mathrm{poly}(k)$ will save a factor of $n$ from the latter complexity of $(n + m) \cdot \mathrm{poly}(k)$.
> For the $\ell_p$ sparse recovery problem, when $p \ge 2$, our algorithm is able to find heavy hitters that are even ``less heavy" than those for $\ell_2$, e.g., imagine a single coordinate $i^*$ has value $n^{1/3}$ while all other coordinates have value $1$. Then $i^*$ must be found by an $\ell_3$ sparse recovery algorithm, but not by an $\ell_2$ sparse recovery algorithm.

---

> > ### Comment · Reviewer_k2z9 · 2023-11-20
> >
> > Thanks for clearing up the proof of Lemma 3.2; it's much clearer now. The integration of OSNAP into the text and the numerical experiments is an important improvement, in my eyes.
> >
> > Unfortunately, the more I look at the numerical experiments and the proposed use cases for the method, I become less convinced this will be a useful algorithmic technique in practice (see below).
> >
> > I understand the authors are coming from a theoretical computer science perspective. As a self-contained work of TCS and divorced from questions of whether the algorithmic problem being solved has practical use, I have no complaints about this work. To me, however, publication of a linear algebraic algorithm in ICLR requires the algorithm to be interesting and useful to practitioners with experiments that appropriately demonstrate its usefulness. My concerns about practical usefulness of the method have only grown the more I've looked into the method and played around with implementing it myself, so I have elected to lower my score.
> >
> > ## Numerical experiments
> >
> > In my original review, I encouraged the authors to use a randomized low-rank approximation algorithm—rather than a full SVD—as the baseline for the numerical comparisons in Table 1. As they have not adopted this suggestion in the most recent revision, I did some of my own experiments.
> >
> > As a test, I generated a random matrix of size 3430 by 6906 and ran four algorithms: the author's method (with OSNAP s=2), Andoni–Nguyen, the Halko–Martinsson–Tropp (HMT) randomized SVD, and a full SVD. I set $m=100$ and $k=10$. Here are the results:
> >
> > - 0.043 seconds for the author's proposal.
> > - 0.071 seconds for Andoni–Ngueyn.
> > - 0.046 seconds for HMT.
> > - 7.22 seconds for a full SVD.
> >
> > But the author's mention that their matrix is sparse, possessing only 353,160 nonzeros. Using this sparsity and representing $A$ using a sparse matrix, I get the following timings:
> >
> > - 0.008 seconds for the author's proposal.
> > - 0.012 seconds for Andoni–Nguyen.
> > - 0.003 seconds for HMT.
> > - 7.31 seconds for a full SVD.
> >
> > Actually computing a full low-rank approximation and the error of this low-rank approximation exactly using the formula $\|A - \hat{A}\|_F^2 = \|A\|_F^2 - \|\hat{A}\|_F^2$ for the HMT low-rank approximation $\hat{A} = QQ^\top A$ is over twice as fast as the proposal on a matrix with the same size and sparsity as in the author's own tests.
> >
> > On the basis of these timings, I believe the author's numerical experiments—as currently designed—are misleading about the purported benefits of the proposed methodology.
> >
> > ## Use Cases
> >
> > >  There are natural settings in which we do not necessarily need to compute a low-rank approximation of all of the input matrices . Instead, we only need to compute an $\alpha$-fraction of them, based on which of them has a good low-rank approximation. In the case when $\alpha$ is small (such as 0.1) and computing an actual low-rank approximation is much more expensive than estimating its cost, our algorithm will achieve a factor of roughly $\alpha$ savings in its resource requirements.
> >
> > I have never encountered or heard of such a scenario occurring in practice. Do the authors have a reference for such a scenario in an applied context?
> >
> > > In the streaming model, the space we need for our algorithm is...
> >
> > I understand that your algorithm has lower space than computing a low-rank approximation. My question is _What use is a low-space algorithm for testing the residual error if, should the low-rank approximation error be small, you'll need higher-space to actually compute the low-rank approximation?_ If I only have $\mathrm{poly}(k)$ space available to me, your algorithm isn't helpful to me. Sure, I can run your algorithm, but I lack enough space to actually do anything useful (i.e., compute a low-rank approximation) after a positive result from your algorithm.
> >
> > The runtime benefits might be a different story, but, as I demonstrate above, your existing experiments don't support a big speedup over just running a randomized low-rank approximation algorithm.
> >
> > > For the $\ell_p$ sparse recovery problem...
> >
> > As a matter of mathematics, I agree with the authors that $\ell_p$ heavy hitters can detect lighter heavy hitters than an $\ell_2$ algorithm does. (This is a point that would be work making in the paper itself, to communicate the purported benefit of the proposed approach to a broad machine learning audience!) But noting that $\ell_p$ heavy hitters can detect lighter heavy hitters, by itself, does not provide justification that the $\ell_p$ residual estimation problem is of interest to machine learning. Does the proposed algorithm lead to concrete benefits on real data sets? Is super-constant space cost of $n^{1-2/p}$ acceptable in practice? To publish an algorithm for this problem in this venue, I think these "applied" questions deserve answers.

---

> > > ### Author Response · Authors · 2023-11-21
> > > **Response to Reviewer k2z9**
> > >
> > > We thank the reviewer for their detailed comments. We have uploaded a new version of our submission.
> > >
> > > The Numerical Experiments.
> > >
> > > We note that the reason we mentioned the time of the SVD is just as a reference time - we did not treat it as a baseline to beat. Indeed, the main point we want to illustrate from the previous version of the experiments is that the sparse sketch we use has a similar accuracy while having a much faster running time than the dense sketch. Following the reviewer's advice, we compare the following two approaches in our new version of the experiments:
> > >
> > > - Randomized SVD: here we use the implementation in the sklearn Library in Python. Note that this method does not support the streaming setting and has a larger space requirement.
> > >
> > > - The algorithm in [CW13]:  which supports the streaming setting and computes a low-rank approximation directly.
> > >
> > > Taking the parameters $m = 50$ and $k = 20$ for the KOS dataset as an example, we have:
> > > - 0.120s  for our approach
> > > - 0.230s for randomized SVD
> > > - 0.660s for the streaming LRA
> > >
> > > Hence, we believe that our algorithm still has a runtime benefit when compared to more baselines for low-rank approximation. (The result of our reports is slightly inconsistent than that of the reviewer's. One possible reason is if the reviewer is using Matlab, then the built-in method is already heavily optimized, which could result in a closer runtime gap).
> > >
> > > On Use Cases.
> > >
> > > > I understand that your algorithm has lower space than computing a low-rank approximation. My question is What use is a low-space algorithm for testing the residual error if, should the low-rank approximation error be small, you'll need higher-space to actually compute the low-rank approximation? "
> > >
> > > One of the reviewer's points seems to be that since the downstream task needs more space if we test a smaller error, then the memory savings is not very useful as we would ultimately need a larger memory. However, note that the data analysis in practice is often conducted simultaneously among different groups of data. Hence, the smaller space requirement in the first stage allows us to do a larger amount of testing among different groups of data during the same period, which results also in a time savings.
> > >
> > > > I have never encountered or heard of such a scenario occurring in practice. Do the authors have a reference for such a scenario in an applied context?
> > >
> > > A typical application of low-rank approximation is to recommendation systems (for more details, see the textbook [Agg16]). In this model, we consider an $m \times n $ preference matrix $A$, where $A_{ij}$ denotes user $i$'s preference for item $j$. Given a matrix preference $A$ with a subset of known entries, the goal is to predict large entries of $A$, representing good recommendations. The method based on low-rank matrix completion works well in practice for this problem.
> > > For this task, it is reasonable to first check whether the current collection of items and users indeed have a good low-rank representation and use them as part of the data in future predictions, which we believe should result in a runtime savings. The $\mathrm{poly}(k)$ space of the algorithm is also beneficial here as usually the prediction is performed simultaneously across different user/item groups, and hence a smaller working space allows for computing more groups at the same time.
> > >
> > > Another potential use of our algorithm is as a sub-routine of more complicated optimization problems. Consider the low-rank tensor decomposition problems in [SGL+20]. At a high level, for each mode of the tensor $X^{(n)}$, we compute a decomposition of $X^{(n)}$, however, if the tail error of $\|X^{(n)} - X^{(n)}_k\|_F^2$ is close to $\|X^{(n)}\|_F^2$, then we could just use an arbitrary orthonormal basis, which will only increase the error by a small factor and we then gain some savings in runtime / memory by not having to explicitly compute a low rank approximation of most flattenings of the input tensor.
> > >
> > > > I agree with the authors that
> > >  heavy hitters can detect lighter heavy hitters than an
> > >  algorithm does. (This is a point that would be work making in the paper itself, to communicate the purported benefit of the proposed approach to a broad machine learning audience!)
> > >
> > >  We have added some discussion in the introduction, and we thank the reviewer for pointing this out.

---

> ### Author Response · Authors · 2023-11-21
> **Response to Reviewer k2z9: continue (due to the max length constraint)**
>
> > “But noting that  heavy hitters can detect lighter heavy hitters, by itself, does not provide justification that the
>  residual estimation problem is of interest to machine learning. Is super-constant space cost of $n^{1-2/p}$
>  acceptable in practice? "
>
> Note that our algorithm indeed also outputs a vector $\hat{x}$ which satisfies the requirements for the $\ell_p$ sparse recovery problem, which was unknown for $p > 2$. We also provide a matching lower bound for the $\ell_p$ recovery problem.  As mentioned, our algorithm for $\ell_p$ is able to find heavy hitters that are even ``less heavy" than those for $\ell_2$, which is beneficial to, e.g., the standard Gaussian white noise model in practice for which the underlying vector is a sparse vector plus Gaussian noise (see. e.g., [WHXF11]), where now we can handle a smaller variance on the Gaussians. The ability to detect the existence of a large signal in compressed sensing is also useful, as we can decide to use more measurements only if we notice the existence of such a signal; otherwise we can use fewer. This would make the sensing adaptive, but this is also a common model in compressed sensing (see, e.g., [IPW12, MN13]).
>
> [CW13]: Kenneth L. Clarkson, David P. Woodruff. Low Rank Approximation and Regression in Input Sparsity Time.
>
> [Agg16]: Charu C. Aggarwal. Recommender Systems: The Textbook.
>
> [CGL+19] Yiming Sun et al. Low-Rank Tucker Approximation of a Tensor from Streaming Data.
>
> [WHXF11]: Ning Wan-zheng et al. The Analysis of Noise Reduction Performance in Compressed Sensing.
>
> [IPW12]: Piotr Indyk, Eric Price, David P. Woodruff. On the Power of Adaptivity in Sparse Recovery
>
> [MN13]: Matthew L. Malloy, Robert D. Nowak .Near-Optimal Adaptive Compressed Sensing.

---

> > ### Comment · Reviewer_k2z9 · 2023-11-22
> >
> > Thanks to the authors for their detailed replies.
> >
> > I remain quite skeptical that either of the proposed methods will be useful in practice (at least for the use cases suggested by the authors), and, as such, I find ICLR a bit of an odd publication venue for this work.
> >
> > That said, I appreciate the paper for its mathematical and TCS contributions. I appreciate the authors for taking the time to thoughtfully respond to my (rather lengthy) reviews, and I believe that the changes made will make the method more robust if it sees use in practice (OSNAP rather than CountSketch) and will be more informative to practitioners about the speedups they can expect (randomized low-rank approximations as baselines).
> >
> > I have raised my score to its original level and wish the authors all the best.

---

### Official Review · Reviewer_Mz93 · 2023-10-26

**Soundness:** 3 good
**Presentation:** 3 good
**Contribution:** 3 good
**Rating:** 8
**Confidence:** 4

**Summary:**

This paper researches the $\textit{residual error estimation}$ problem. To be specific, for a matrix $A$, one estimates $|| A - A_k ||_F$ and judges if it is worthwhile to compute a low-rank approximation $\widehat{A}$ such that $|| A - \widehat{A} ||_F \le (1 + \varepsilon) || A - A_k ||_F$, where $A_k$ is the best rank-$k$ approximation to $A$; for a vector $x$, one estimates $|| x - x_k ||_p$ and determines whether to compute a $k$-sparse vector $\widehat{x}$ such that $|| x - \widehat{x} ||_p \le (1+\varepsilon) || x - x_k ||_p$, where $x_k$ is the best $k$-sparse approximation to $x$.
Given a matrix $A$, for the estimator $|| SAT - [SAT]_k ||_F$ of the residual error $|| A - A_k ||_F$, this paper obtains the lower bound and upper bound of the dimensionality $s, t = \Theta(k / \varepsilon^2)$, which improves the upper bound $O(k / \varepsilon^3)$ in [Andoni and Nguyen'13].
Given a vector $x$, for the estimator of $|| x - x_k ||_p$ with $p > 2$, this paper gives the upper bound $\widetilde{O}(\varepsilon^{-2p/(p-1)} k^{2/p} n^{1-2/p})$ for space. This result can be extended to the $\ell_p$ sparse recovery problem, in which the upper bound almost matches the obtained lower bound $\Omega(k^{2/p} n^{1-2/p})$ when taking $\varepsilon$ as a constant.

**Strengths:**

(1) For low-rank residual error estimation, this paper gives the lower bound $\Omega( k / \varepsilon^2 )$ for the dimensionality of the sketching matrices $S$ and $T$, and the upper bound $O(k / \varepsilon^2)$, which matches the lower bound and improves the upper bound $O(k / \varepsilon^3)$ in [Andoni and Nguyen'13].
(2) For residual error estimation of vector $\ell_p$ norm with $p > 2$, this paper gives an upper bound $\widetilde{O}(\varepsilon^{-2p/(p-1)} k^{2/p} n^{1-2/p})$ for the dimensinality of the sketching matrix. Also, this upper bound can be extended to $\ell_p$ sparse recovery problem and almost matches the obtained lower bound $\Omega(k^{2/p} n^{1-2/p})$ when $\varepsilon$ is a constant.
(3) The experimental results indicate that the proposed algorithm has a comparable error $\varepsilon$ but at least ten times faster than the existing method [Andoni and Nguyen'13].
(4) This paper is well-written and easy to understand.

**Weaknesses:**

In the experiments, this paper only proves the efficiency of the proposed algorithm for low-rank approximation. Can this paper supplement experiments for the residual error estimation of vector norms?

**Questions:**

See Weaknesses.

---

> ### Author Response · Authors · 2023-11-17
> **Response to Reviewer Mz93**
>
> We thank the reviewer for their thoughtful comments. We can provide additional experiments in the next version of our paper.

---

> > ### Comment · Reviewer_Mz93 · 2023-11-20
> >
> > Thank you for your feedback. I will retain my score.

---

### Official Review · Reviewer_QAtz · 2023-11-01

**Soundness:** 3 good
**Presentation:** 3 good
**Contribution:** 3 good
**Rating:** 8
**Confidence:** 3

**Summary:**

This paper studies the problem of estimating the residual error using linear sketching. Specifically it focusses on the following two problems:

1) For any matrix A, the task is to estimate $||A-A_k||_F$ within a $(1+\epsilon)$-factor where $A_k$ is the best rank-k approximation to $A \in \mathbb{R}^{n \times d}$. For this problem, it is shown a bilinear sketch of the form $SAT$ where the sketching matrices $S \in \mathbb{R}^{O(k/\epsilon^2) \times n}$ and $T \in \mathbb{R}^{d \times O(k/\epsilon^2)}$ can be sparse countsketch matrices with a single $\pm 1$ in their columns. This gives a sketch of size $O(k^2/\epsilon^4)$. A lower bound is given which shows S and T must have $O(k/\epsilon^2)$ rows and columns respectively for any algorithm which takes as input $S,T,SAT$ and outputs a $(1+\epsilon)$ residual ran-k approximation. This shows the upper bound is tight for bilinear sketches.

2) The next problem is estimating $\| x-x_k\|_p$ upto $(1 \pm \epsilon)$ residual error for any vector $x \in \mathbb{R}^{n}$ which is updated in the streaming model, for any $l_p$ norm with $p>2$ and where $x_k$ is the the vector containing the top-k elements of $x$. The algorithm uses countsketch to find the indices of top-k elements of $x$ approximately and parallely uses a known $F_p$ frequency estimation procedure for these coordinates and subtracts the approximate value of these coordinates to get the residual error. This yields an upper bound of $\tilde{O}(\epsilon^{-2p/(1-p)}k^{2/p} n^{1-2/p} )$ and also gives a $k$-sparse approximation for $x$. A lower bound of $\Omega(k^{2/p} n^{1-2/p})$ is also shown for the $k$-sparse recovery problem.

Some experiments are performed with the sparse sketching matrices.

**Strengths:**

1.  For the residual matrix norm estimation problem, the paper presents the bilinear sketches which are optimal in size. Moreover, since the sketching matrices can be sparse, they can be computed are pretty fast. Previous results for this problem used dense sketching matrices and had a larger upper bound of $O(k^2/\epsilon^6)$ on the size of the sketch. The proof for the sketching algorithm is pretty straightforward from results on projection cost preserving sketches. The lower bound proof follow via minimum singular value separation results for random Gaussian matrices.

2. For the residual vector norm recovery problem, the algorithm also outputs a $k-sparse$ approximation in $l_p$ norm which is optimal upto polylog $n$ factors (though the dependence on $\epsilon$ is unclear). The lower bound proof follows from reducing the gap infinity problem to the $l_p$ sparse recovery problem.

3. Though some of the results are pretty straightforward from prior results (for example, the bilinear sketch upper bound follows directly from the PCP sketch properties), they are an interesting addition to the sketching literature.

4. The paper is mostly well written and easy to follow apart from some typos etc. (see weakness and questions)

**Weaknesses:**

1) It is unclear how good the upper for the residual norm estimation problem as compared to the optimal. The given lower bound applies to the sparse recovery problem. But this is a harder problem than the norm residual error estimation problem. Some discussion on the actual lower bound for the norm estimation problem might make things more clear for the reader. Also, though the upper bound for the sparse recovery problem is optimal in terms of n and k, it is unclear on what the correct dependence on $\epsilon$ is.

3) There are some typos etc. in the paper which should be corrected:
      1) In definition 3.4, $S$ should be $s \times n$
      2) The proof of theorem 3.7 should refer to lemma 3.5 instead of 3.4.

**Questions:**

1) Can anything be said regarding how good $SAT$ is as compared to $A_k$, the best rank-k approximation to A?

2) I'm confused about the the proof of Lemma 4.3. It seems we should have $(I \cap J) \cup T_1 \cup \ldots T_{m+1} \subseteq I$ instead of exactly being equal to $I$ as it could happen that $i \in I \backslash J$ and  $x_i > || x_{-k}||_p $  in which case $i$ wouldn't belong to any of

the sets $ T_1, \ldots T_{m+1}$? It seems instead of decomposing I, decomposing J as $ J= (I \cap J) \cup T_1 \cup \ldots T_{m+1}$ where each $T_i$ constains elements of $J$ not in $I$ in the given interval and then doing the analysis should given the same result?

Remark: I have not checked whether Lemma 5.2 of Price and Woodruff, 2011 indeed generalizes to p>2. This is used in Lemma 4.10.

---

> ### Author Response · Authors · 2023-11-17
> **Response to Reviewer QAtz**
>
> We thank the reviewer for their thoughtful and detailed comments. We have fixed the typos mentioned and uploaded a new version of the paper. We agree that there is still a bit of a gap between the bounds for the sparse recovery and residual norm estimation problems we study in this paper, and we treat this as an interesting future direction.
>
> - Question 1: Note that the size of $SAT$ is different from that of $A_k$, so they are not comparable as matrices. We have shown though in Theorem 3.7 that the residual error of $SAT$ is a good approximation to the residual error of using $A_k$ for $A$. Note though that $SAT$ is a much smaller sketch of $A$ and so gives a low memory way of approximating the residual that one could have found by using $A_k$. Also, in the low-rank approximation problem, using matrices $SA, AR, SAT$, we can more efficiently (than computing $A_k$ directly) compute two matrices $U$ and $V$ such that
> $
> ||A - UV||_F^2 \le (1 + \epsilon)||A - A_k||_F^2 \;.
> $
> (See the survey [1] for more details.)
>
> - Question 2: We thank the reviewer for pointing this out. In fact, one can show that $\vert x_i\vert ^p \leq 2 \Vert x_{-k}\Vert_p^p$ for all $x_i \in I \setminus J$. Defining a new magnitude level for coordinates between $\Vert x_{-k} \Vert_p^p$ and $2\Vert x_{-k} \Vert_p^p$ is enough and the remainder of the proof is almost the same. The updated proof has been included in the new version which we have uploaded.
>
> [1] David P. Woodruff. Sketching as a Tool for Numerical Linear Algebra.

---

> > ### Comment · Reviewer_QAtz · 2023-11-21
> >
> > Thanks for your response and for addressing my concerns in the new version. My original score remains unchanged.

---

### Official Review · Reviewer_FqHJ · 2023-11-05

**Soundness:** 3 good
**Presentation:** 4 excellent
**Contribution:** 3 good
**Rating:** 6
**Confidence:** 3

**Summary:**

This paper studies the (approximate) estimation of residual error of rank-k approximation of matrix and k-sparse recovery of vector using linear sketches. For the matrix case (Frobenius norm), the main result is a bilinear sketch of size $O(k^2/\epsilon^4)$ by combining the count sketch and JL sketch; this improves upon the previous best result $O(k^2/\epsilon^6)$ [Andoni-Nguyen, SODA'13]. The authors also provide a matching lower bound. For the $k$-sparse recovery under $\ell_p$ norm for $p>2$, the author show an upper bound of $\tilde{O}(k^{2/p}n^{1-2/p}$. This is done by two sketches, where one is used to find heavy hitters, and another to do a $F_p$ estimation.

**Strengths:**

I think this paper is technically solid and solves an interesting open problem. The high-level explanation provided in the paper are helpful to the readers.

**Weaknesses:**

1. For the matrix case, the improvement compared to Andoni and Nguyen seem a bit incremental.
2. While I think it makes sense to estimate the residual error of matrix approximations, I think there is little use to estimate the residual error of sparse recovery under $\ell_p$ norm.

**Questions:**

See weakness.

---

> ### Author Response · Authors · 2023-11-17
> **Response to Reviewer FqHJ**
>
> We thank the reviewer for their thoughtful comments.
>
> - On our improvement over Andoni and Nguyen: we improve the upper bound of Andoni and Nguyen from $O(k^2/\epsilon^6)$ to $O(k^2/\epsilon^4)$.
> Note that the $O(k^2/\epsilon^6)$ bound in their work was also only shown to hold for Gaussian matrices. In our work, we give a very different analysis than that in Andoni and Nguyen, and instead directly show that any sketching matrix with the so-called projection cost-preserving (PCP) property suffices, which allows us to directly use extremely sparse sketching matrices with a much faster running time, and also allows us to improve the bound to $O(k^2/\epsilon^4)$ using advances in PCPs. We also show an $\Omega(k^2/\epsilon^4)$ lower bound, which shows that our upper bound is optimal up to a constant factor.
>
> - For the $\ell_p$ sparse recovery problem, when $p \ge 2$, our algorithm is able to find heavy hitters that are even ``less heavy" than those for $\ell_2$, e.g., imagine a single coordinate $i^*$ has value $n^{1/3}$ while all other coordinates have value $1$. Then $i^*$ must be found by an $\ell_3$ sparse recovery algorithm, but not by an $\ell_2$ sparse recovery algorithm.

---

### Meta-Review · Area_Chair_EFKo · 2023-12-05

**Metareview:**

The paper considers the problem of estimating the residual error when using a low rank approximation for a matrix, or using a sparse approximation for a vector. The paper significantly improves the previous results for residual error estimation in both the memory bound and the choices of the sketching matrix, getting tight bound for this case, and it develops new algorithms for the vector case. All reviewers are supportive of the theoretical contribution of the paper.

On the other hand, one reviewer is concerned that the runtime/memory improvement as part of a larger application might not be significant, which limit the potential scenarios where it is worthwhile to perform the estimation before computing the estimation. In particular, the cost of estimation is within a logarithmic factor of computing the actual factorization in theory and in practice, the difference seems negligible for some experiments done by the reviewer. Other reviewers also note that some proof techniques are straightforward, and the lower bound for the vector case is for a different and harder problem so it is unclear if it is a good evidence for the hardness of error estimation.

**Justification For Why Not Higher Score:**

Given two high scores, it can be accepted as a spotlight if there is room. The contribution is definitely more theoretical than immediately practical so the decision here depends on the fit with the goal of the conference.

**Justification For Why Not Lower Score:**

All reviewers are supportive of the theoretical contribution of the paper, with several high scores so the paper definitely merits acceptance.

---

### Decision · Program_Chairs · 2024-01-16

Accept (poster)